# Automatic Classification Service System for Citrus Pest Recognition Based on Deep Learning

**DOI:** 10.3390/s22228911

**Published:** 2022-11-18

**Authors:** Saebom Lee, Gyuho Choi, Hyun-Cheol Park, Chang Choi

**Affiliations:** 1Department of Computer Engineering, Gachon University, Sujeong-gu, Seongnam-si 461-701, Gyeonggi-do, Republic of Korea; 2Department of AI Software, Gachon University, Sujeong-gu, Seongnam-si 461-701, Gyeonggi-do, Republic of Korea

**Keywords:** agriculture, citrus disease classification, deep learning, web application

## Abstract

Plant diseases are a major cause of reduction in agricultural output, which leads to severe economic losses and unstable food supply. The citrus plant is an economically important fruit crop grown and produced worldwide. However, citrus plants are easily affected by various factors, such as climate change, pests, and diseases, resulting in reduced yield and quality. Advances in computer vision in recent years have been widely used for plant disease detection and classification, providing opportunities for early disease detection, and resulting in improvements in agriculture. Particularly, the early and accurate detection of citrus diseases, which are vulnerable to pests, is very important to prevent the spread of pests and reduce crop damage. Research on citrus pest disease is ongoing, but it is difficult to apply research results to cultivation owing to a lack of datasets for research and limited types of pests. In this study, we built a dataset by self-collecting a total of 20,000 citrus pest images, including fruits and leaves, from actual cultivation sites. The constructed dataset was trained, verified, and tested using a model that had undergone five transfer learning steps. All models used in the experiment had an average accuracy of 97% or more and an average f1 score of 96% or more. We built a web application server using the EfficientNet-b0 model, which exhibited the best performance among the five learning models. The built web application tested citrus pest disease using image samples collected from websites other than the self-collected image samples and prepared data, and both samples correctly classified the disease. The citrus pest automatic diagnosis web system using the model proposed in this study plays a useful auxiliary role in recognizing and classifying citrus diseases. This can, in turn, help improve the overall quality of citrus fruits.

## 1. Introduction

Agriculture is a pivotal component of the global economy. Through digital innovations and technological development, the agriculture industry is developing into a fast-growing future and is a core industry of the Fourth Industrial Revolution. However, climate change, growing conditions in arable land, old and new plant diseases, and pests are important obstacles to growing crops, despite advances in agricultural technology. In particular, according to the 21st Century Guidebook to Fungi [1], approximately 16% of crops worldwide are losing value due to plant pests, and not only wheat and rice, but also fruit crops are severely damaged by pests. Fruits are crops grown and traded around the world, accounting for a huge proportion of the agriculture industry. In particular, the most consumed and traded fruit crop is citrus [2]. It is generally produced and cultivated in approximately 140 countries, and includes varieties such as mandarin, orange, citron, and lime, depending on the location and climate of the plantation [3]. The citrus yield is easily affected by the weather, and the plant is very vulnerable to plant pests. For example, in citrus fruits, diseases such as melanin, canker, and scab often occur after rain. As a result, citrus production and cultivation cause significant economic losses in all growing regions of the world and continue to be threatened by many factors, such as pathogens and pests [4]. The state of Florida in the USA spent USD 1 billion on the eradication of citrus pests for ten years from 1995 to 2005, and the state of Queensland in Australia spent AUD 18.5 million for the same goal [5]. In addition, in Jeju, Korea, tangerine crops are damaged by various pests, such as citrus canker and *Panonychus citri*, every year. To prevent pests, a sustainable and effective strategy has been proposed to improve varieties resistant to plant pests [6]. New variety improvement programs through biotechnology tools have been developed and improved, but the cultivation of citrus varieties that are highly resistant to the same disease remains a challenge due to laboratories and experimental conditions [7] different from the actual plantation [8]. To date, there is no successful approach to eradicating citrus pests, and the only fundamental way to prevent the spread of the disease and minimize the effects of infection is the removal of infected plants from pathogens. Numerous methods have been applied and introduced to address problems related to plant pests, but early detection and accurate diagnosis of diseases must be prioritized to reduce the spread of diseases and crop damage. Predicting and detecting diseases early, monitoring plant health, and applying control measures accordingly is critical to sustainable agriculture. Therefore, there is a need for the latest AI-based tools that can predict the prognosis of diseases by detecting pests in advance.

In recent years, research on plant pests has been conducted, focusing on various machine-learning and deep-learning technologies, along with the increase in computing resources due to the development of computer systems [9,10,11]. Since machine learning methods operate based on user-defined functions, image feature values may be missed in image classification. CNN-based deep learning technology has been used in recent plant disease diagnosis research to solve this problem. Deep learning models have a longer runtime for training, but a shorter runtime than other methods. They also provide more accurate results than machine-learning methods by automatically extracting image features from raw images/data [12]. Therefore, the deep learning-based plant disease classification method is the most promising computing method in modern agriculture because it derives better accuracy than machine learning. In many studies, deep-learning models are used to identify pests in various parts of the plant, such as citrus fruits, leaves, and stems [13,14,15,16]. Computer vision models require training on large image datasets to increase the accuracy of classification and detection. However, the publicly available citrus pest dataset is very limited and contains low-resolution disease sample images. The datasets used in the current study are limited in the variety of pests by collecting images from Kaggle and Citrus Image Gallery Dataset, and the dataset is relatively small [17,18,19], containing 5000 images. In addition, the citrus pest disease classification study [20] using a dataset obtained in the laboratory achieved high performance but is not a dataset obtained from the actual plantation, so it is difficult to project the research results directly to the plantation.

In order to solve the problem of datasets in existing studies, this paper builds big data by collecting citrus pest images on its own from actual citrus plantations. The constructed dataset trains five deep learning models to classify citrus pests and diseases. The five models extract weights derived from the highest f1 score through training, and the extracted weights are used in the test dataset to evaluate the performance of the proposed model. The best model is applied to the web application system that informs the type of disease when the user inserts an image. The proposed deep learning-based automatic disease classification system for citrus has achieved excellent performance, and it can help to quickly and easily identify pests in the agricultural industry, where early diagnosis of pests and pests is difficult.

In summary, the significant contributions of this study are as follows:We have developed a novel citrus pest dataset comprising six disease detection classes. The constructed dataset consists citrus images that are either infected or non-infected by pests in Jeju Island, South Korea, in 2021. The constructed dataset provides a total of 20,000 high-quality images with a resolution of 1920 × 1090. Currently, Citrus Open Datasets are either low resolution or paid. We published the datasets used in the study free of charge https://github.com/LeeSaeBom/citrus (accessed on 19 August 2022). A detailed description of the dataset is provided in Section 5.We use EfficientNet and ViT models, which are the latest algorithms in this area, including VGGNet, ResNet, and DenseNet models, which are commonly used for the classification and detection of plant pests and diseases [21,22,23]. All five models can use the pre-training method, and high accuracy and f1 score derivation are possible. VGGNet, ResNet, DenseNet and EfficientNet models can extract local features of the feature map using a convolution layer, and the ViT model uses a transformer, so global features of the feature map can be extracted.Application development is required to automate the classification of various diseases. The web application server has the advantage that it can be accessed from anywhere in the world, as long as the Internet is available. In most citrus cultivation sites, workers manually determine the presence or absence of pests and classify disease types. It is difficult for non-professional workers to quickly determine the type of pest. Based on these problems, we developed our own web application system, and non-professional workers can use it to easily determine the pests and diseases.

The rest of this paper is organized as follows. The related work is mentioned in Section 2. The Network Architecture presented in Section 3. The Method is mentioned in Section 4. The experimental procedure and results are discussed in Section 5, and finally, the conclusion and future work are addressed in Section 6.

## 2. Related Works

In the past, researchers used statistics, machine learning, and deep learning approaches to classify and detect plant pests. Statistics and machine learning detect plant diseases based on features of self-produced images. Traditional machine-learning methods tend to rely heavily on user-defined image features, so important features of the image may be missed. This, along with degraded model performance, leads to difficulties in detecting immediate plant pests. To overcome this problem, a deep-learning-based approach that passes through the layers and uses the image features determined from the layers is used to solve various orchard disease classification problems.

### 2.1. A Study on the Detection of Fruit Crop Disease

Before classifying citrus disease, classification studies on various crops were conducted. Wang et al. [24] proceeded with disease classification for tomatoes using an artificial neural network (ANN). Singh and Misra [25] et al. also used ANN to detect disease in bananas. Sankaran et al. [26] performed citrus disease classification using k-nearest neighbor (KNN). In addition, various plant and crop disease classification studies using plant village, an open dataset, are continuously being conducted [27].

### 2.2. A Study on Fruit Crop Disease Based on Machine Learning

As computing speed improves, studies have been conducted to classify diseases of various fruit crops, focusing on machine-learning approaches. Among machine-learning approaches, support vector machine (SVM) is a widely known method for detecting diseases of citrus [28] and tomatoes [29]. Iniyan et al. [30] combined SVM and ANN to obtain high accuracy in orchard disease detection. Another representative method among machine-learning-based infected plant recognition research is to use principal component analysis (PCA) [31]. The PCA method has the advantage of finding components that are judged to explain the given data well. However, as statistics and machine learning detect plant diseases based on self-generated image features, important information in the images may be lost. This, along with poor model performance, leads to difficulties in immediately detecting plant pests. Deep learning methods solve problems in machine learning by going through layers and using features of an image determined by the layers. The downside of deep learning models is that they take longer than machine learning methods. However, with the advancement of computing systems, the training time of deep learning-based models is decreasing. Therefore, deep learning techniques are more effective than machine learning in plant disease recognition research. In order to solve this problem over the past 5 years, research combining machine learning and deep learning has appeared, and deep network learning has become possible with the development of [32,33] GPUs, and deep-learning-based crop disease classification research is in progress [34].

### 2.3. A Study on Fruit Crop Diseases Based on Deep Learning

Fruit crop disease recognition research is a study of datasets including various types of fruit, such as bananas and tomatoes. As labeling data increased and imaging techniques developed, datasets for single breeds increased. This part focuses on disease awareness research on citrus cultivars. Research on the classification of citrus and pest diseases has been conducted in various ways. Among them, light convolutional neural network, multi-class support vector machine (M-SVM), pyramid histogram of oriented gradients (PHOG), ensemble boosted tree [35], linear discriminant analysis [36], convolutional neural networks(CNN) [37], very deep convolutional network (VGGNet) [38] for large-scale image recognition, and other deep-learning and machine-learning methods were combined to train the model and classify orchard diseases.

Hossain et al. [39] proposed a deep-learning model capable of classifying several fruit crops, including citrus, in a variety of commercial environments, and used two CNN models. The first used the light CNN model, and the second used the VGG16 model to compare the performance with other models. Nasir et al. [40] conducted a deep-learning-based classification of fruit-crop-related diseases and used a method that combines a pre-learned VGG19 model with a PHOG model. After combining the feature vectors obtained through VGG19 and the PHOG model, minimum redundancy maximum relevancy was performed to perform feature selection. Using the obtained features, the accuracy tests were conducted using various classifiers. Syed Ab Rahman et al. [19] detected citrus disease using a two-step deep convolutional neural network based on Faster R-CNN. The network structure is a feature extractor, region proposal network (RPN), region of interest (ROI) pooling, and it consists of four components of a classifier. This study has the advantage of fast training speed and memory saving. Khanramaki et al. [41] conducted a self-constructed citrus pest dataset recognition study using feature-level diversity, data-level diversity, and classifier-level diversity methods. The researchers collected images of three citrus pests. The collected images were data preprocessed in feature-level diversity. In data-level diversity, a bootstrap strategy was used. The size of each bootstrap was matched to the number of training samples, and four bootstrapping results were generated for each basic CNN. In classifier-level diversity, instead of searching for a super-high-level CNN model, a more contextual ensemble classifier was constructed by converging CNN models with common elements compared to the basic CNN. The CNN models used in the study are AlexNet, VGG16, ResNet50, and InceptionResNetV2.

Several studies have shown how to classify citrus pests and fruit crops using CNN and VGGNet. CNN and VGGNet have a high network speed, but they have low accuracy owing to the plain model structure. Therefore, in this study, while utilizing the VGGNet used in the previous research, deep residual learning for image recognition (ResNet) [42], and densely connected convolutional networks (DenseNet) [43] models with CNN-based multi branch structure and EfficientNet: rethinking model scaling for convolutional neural networks (EfficientNet) [44], which propose the compound scaling method, were added. In addition, for comparison and evaluation with CNN-based models, a vision transformer (ViT) [45] with a transformer structure was added to classify citrus pest diseases.

## 3. Network Architecture

The proposed network architecture for this study is to extract the model weights and perform model testing and web applications using the extracted weights. Figure 1 shows a block diagram of the proposed model in a high-level view. The network architecture is mainly divided into the network structure and web application module. The network structure step primarily consists of data transformation, model training, validation, and testing. In the data transformation step [46], model training was conducted after image resizing at 224 × 224 and stochastically rotating the image vertically or horizontally. The detailed data transformation process is described in Section 5. The models used by the network structure were VGGNet, ResNet, DenseNet, EfficientNet, and ViT models, were all pre-trained.

A detailed description of the model is provided in Section 4. The transformed images enter the input layer of each model through the layer of each model and reach the output layer. In this study, the output layer was implemented to derive six diseases by changing the output to six classes in the last step. During the training and validation processes, the models were trained according to a set epoch. In this process, the highest accuracy of the five models and the model weights were extracted with the F1 score. Among them, the model weight that derived the highest F1 score proceeded to the model testing step. In the testing process, the test was performed using the test dataset with the same epoch as in training and validation. Subsequently, the models determined to be suitable were transmitted to the web application module and became the standard of the classification model. The web application module classified the citrus pest disease using the model preferred by the user from among the models delivered from the network structure in the web server designed using Python Flask, and the characteristics and control methods of the pests could be obtained. In this step, when a user connected to an application sends an API request for citrus image classification, it is designed to respond to the classification result value. The detailed process for the web application module is explained in Section 5.

## 4. Methods

In this study, VGGNet16 [38], ResNet50 [42], DenseNet161 [43], EfficientNet_b0 [44], and ViT_b_16 [45] were trained on ImageNet. VGGNet, ResNet, DenseNet, and EfficientNet are algorithms derived from CNN, which trained the network deeper and reduced the error rate. In all four models, when input images were obtained, feature maps were extracted for each layer step through the convolution layer, and the final features were extracted from the last layer through classifier yθ to classify classes. ViT is a model that uses the transformer [47] structure, which is often used in natural language processing (NLP), performs classification using image patches, and demonstrates good performance in large-scale image datasets. In this study, after using a pre-trained model that can derive high classification results in a short time by pre-learning 1000 classes for all five models, the loss of parameter yθ is minimized by applying the cross-entropy loss among the objective functions. Subsequently, for model optimization, the Adam optimizer [48] combined with the RMSProp and momentum method was used, and CosineAnnealingLR [49], a method that finds the optimum point while the learning rate oscillates, was used to train the model and perform citrus pest disease classification.

### 4.1. VGGNet

VGGNet [38] is a model developed by VGG, a research team at Oxford University, and was a runner-up in the ImageNet Large Scale Visual Recognition Challenge (ILSVRC) in 2014. The VGGNet is divided into VGG16 and VGG19 according to the number of convolution layers. VGGNet has a very small convolution filter of 3 × 3 as a structure designed to determine how deepening of the network affects performance. VGGNet receives a 224 × 224 × 3 image as an input and generates feature maps from the first layer (conv1_1) to the 13th layer (conv5_3) through 3 × 3 convolution filters and the ReLU function. Layers 14–16 pass through the fully connected layer to flatten the feature maps. The final output values are a model that consists of 1000 neurons using the softmax function and classifies 1000 image classes.

### 4.2. ResNet

ResNet [42] is a model that won the 2015 ILSVRC and is an algorithm developed by Microsoft. The ResNet model lowered the top five errors by stacking 152 layers seven times deeper than the GoogLeNet model, which won the 2014 ILSVRC. In general, the deeper the network, the better the performance. However, after creating a 20-layer network and a 56-layer network with convolution layers and fully connected layers, respectively, the performance was tested, and the 56-layer network with a deeper structure showed worse performance than the 20-layer network. Therefore, ResNet presents a residual learning framework, which is a novel method used to facilitate the training of much deeper networks. Residual learning reconstructs the layer by learning the residual function by referring to the layer input, instead of learning the unreferenced function. This residual network is easy to optimize, and high accuracy can be obtained, even in deep networks.

### 4.3. DenseNet

DenseNet [43] is a model with a higher performance and fewer parameters than ResNet and Pre-Activation ResNet. DenseNet uses a method of connecting the feature maps of all layers and connecting the feature maps of the previous layer to the feature maps of all subsequent layers. In the case of the ResNet model, the previous layer is added and connected through the residual block, while in DenseNet, all layers are connected through the concatenation method. Therefore, the feature map size is the same when connecting layers, and a very small value is used for the number of feature map channels in each layer, considering that the number of channels is extremely large. The DenseNet model shows that convolutional networks can train much deeper, more accurately, and efficiently through short connections between layers close to the input and those close to the output. This condition alleviates the problem of gradient loss, strengthens feature propagation, encourages feature reuse, and significantly reduces the number of parameters to achieve good performance with less computational cost than previous models.

### 4.4. EfficientNet

CNN is generally developed from limited resources and then expanded to achieve high accuracy when more resources are available. Optimal performance and efficiency were not obtained when increasing the accuracy of the model because the network depth, width, and resolution of input images were manually adjusted. EfficientNet [44] systematically studies model scaling and demonstrates good performance when balancing the depth, width, and resolution of the network. Experiments showed that depth, weight, and resolution have constant relationships. Therefore, the depth, width, and resolution dimensions of all networks can be applied to MobileNet [50] and ResNet using a new scaling method compound coefficient, which is simple and highly efficient, to achieve much better accuracy and efficiency than the previous CNN.

### 4.5. ViT

ViT [45] is a model that applies a transformer, which is used in the field of NLP, for image classification without the use of conventional CNNs with minor variations. The ViT splits an image into patches, passes these patches into linear embedding, and uses them as inputs to the transformer. Similar to the transformer of NLP, uses an embedding combined with learnable positional embedding and element-wise sum and maintains the encoder–decoder structure, which is the existing seq2seq structure. After pre-training using a multi-layer perceptron with one hidden layer for the transformer encoder output, a randomly initialized linear layer is used for fine-tuning, and the pre-trained learned positional embedding adjusts the positional embedding of the image to a different resolution during fine-tuning. In addition, at the beginning of learning, position embedding does not provide any information regarding the 2D position of the patch, and the spatial relationship between the patches is learned from the beginning. The ViT has the advantage of being strong in training a very large dataset with 100B parameters, and with an increase in the size of the dataset, the performance does not saturate, even if the model is expanded. Hence, this model that improves computational efficiency and scalability.

### 4.6. Comparison of Five Models

Each of the five models used in the experiment has a different model structure, and even with the same model, the size of the parameters generated according to the number of layers is different. Table 1 shows the total parameter size of each model.

VGGNet16 is the model with the largest parameter size. As VGGNet16 learns 13 conv layers, the model deepens, and the parameter size becomes very large. After that, dropout is applied to the remaining three layers to prevent overfitting, and some output values of the layers are randomly excluded. As a result, 14,714,688 parameters are generated while passing through 13 conv layers, and 119,570,436 parameter sizes are generated through the remaining three layers. The total parameter size generated in the VGGNet16 model is 134,285,126.

The ResNet50 model consists of 50 deep convolutional neural networks. ResNet50 has a larger number of layers than VGGNet16, so the number of generated parameters is likely to be larger, but it has a smaller parameter size, using residual blocks rather than plane layer structures. The formula of the residual block is shown in (1). The H(x) formula uses the conv layer, batch normalization, and relu function operations. The parameter size generated in the ResNet50 model is 23,520,326 which is 110,764,800, smaller than that of VGGNet16.
(1)$$\begin{array}{c} H(x)=F(x)+x \end{array}$$

The DenseNet161 model has a structure with 161 layers consisting of 156 dense layer blocks in each sub-block and the upper five conv layers. In ResNet, the input value is added to the output value and the gradient flow is transmitted directly. Since these residual connections are combined by addition, information flow in the neural network may be delayed. The DenseNet model improves information flow with a dense block structure that connects the previous layer directly to all the next layers. The formula for the dense block is shown in (2). The dense block consists of 3 × 3 conv layers, batch normalization, and relu function. The parameter size of DenseNet161 for the dataset used in the experiment is 26,485,254. The parameter size is 2,937,928 larger than that of the ResNet model, but the difference is insignificant.
(2)$$\begin{array}{c} X_{l}=H_{l}\left(\left[x_{0},x_{1},...,x_{l-1}\right]\right) \end{array}$$

The ViT model is a transformer structure and is based on the configuration used for BERT. The ViT model is largely divided into ViT_Base, ViT_Large and ViT_Huge. Of the three models, the one used in the experiment was the ViT_b_16 model. ViT_b_16 consists of 12 layers, 768 hidden sizes, 3072 MLP size, and 12 heads. Each layer parameter is 7,087,872, and the total number of parameters is 85,803,270.

The EfficientNet_b0 model consists of 237 layers. EfficientNet_b0 uses the compound scaling method as a way to achieve maximum efficiency with limited resources. The formula for the compound scaling is shown in (3). Because the amount of calculation is proportional to the depth and proportional to the square of the remaining two variables, the movement of the variables can be determined in the same proportion as in Formula (3). These three variables—depth, width, and resolution are closely related. The compound scaling method shows fast inference times.
(3)$$\begin{array}{cc} & depth:d=\alpha^{\phi },\;width:w=\beta^{\phi },\;resolution:r=\gamma^{\phi }\\ & s.t.\;\alpha \cdot \beta^{2}\cdot \gamma^{2}\cdot \approx 2,\alpha \geq 1,\beta \geq 1,\gamma \geq 1 \end{array}$$

As shown in Table 1, the parameter size of EfficientNet_b0 is 4,015,234, which has the smallest parameter size. Among the five models, we show that the EfficientNet_b0 model proposed in this study is highly efficient in computational time.

## 5. Experiments

### 5.1. Citrus Disease Images and Datasets

Among fruit crops, citrus is very sensitive to the environment. The most common citrus pest in subtropical climates is the citrus bacterial canker (CBC) [51], and the most common citrus pest in Asia is citrus huanglongbing (HLB) [52]. The causative agent of CBC is Xanthomonas citri subsp (XCC). The XCC bacteria occur in the majority of citrus fruits, including grapefruit, lemon, lime, and sweet orange. This disease has symptoms, such as early leaf and fruit dropping, dryness of small branches, brown spots, and small blister-like lesions on leaves, fruits, and stems that start small and expand as the disease progresses. CBC is a citrus pest mainly in Australia, the USA, and South American countries, but it has recently been observed in Asia and is emerging as a worldwide citrus disease.

HLB is a pathogen with a destructive effect on the citrus industry and is classified as the most serious citrus pathogen. This pest is caused by Asian citrus psyllid, and the causative bacterium is Candidatus liberibacter spp. The HLB disease is transmitted by an insect vector, and there is no cure for citrus trees affected by the disease. Hence, prevention and felling of infected trees are the only ways to stop the spread of diseases. Although it is the same tangerine, various diseases and pests appear, depending on the growing region and environment, and their types are also different. Therefore, to study citrus pest diagnosis and classification models applicable to agricultural environments in several regions beyond one region, it is necessary to establish a large dataset that includes various citrus pests.

Many of the current studies [17,18,19] mainly used images obtained from open datasets such as the Kaggle and citrus Disease Image Gallery Datasets. Both datasets are similar to midstream citrus pests mainly found in Australia and the USA, and the majority of citrus-related image classification studies used subtropical climate citrus species. In this study, among the citrus obtained in Asia, citrus trees that have similar pests to CBC and HLB and can collect new pest images were investigated. CBC pests were found in citrus unshiu, red mites, and aphids, which kill trees in the same way as HLB. Red mites cause Panonychus citri disease [53], and aphid is a pest causing Toxoptera citricida disease [54], a type of pest that has not been examined in previous studies.

In this study, images of pest-free citrus fruits and pest-infected images (including fruits and leaves) were acquired from a citrus farm in Jeju, Korea, for the year 2021. There is a risk of noise problems with image datasets obtained from actual plantations. To avoid the noise problem, we shot so that the subject was visible under various conditions. First, the exact screen composition and ratio were set so that the entire subject came out. After that, the height, angle, distance, and lighting distance were adjusted for each pest type for consistent image quality. Additionally, only a single breed was photographed on a white background, and the photograph was taken so that the subject’s shaking, shading, and light reflection did not occur at the time of the shooting. Finally, the pictures were taken so that the unique characteristics of each breed, such as color and pattern, were clearly visible. High-quality images were collected using this method. Therefore, the rate of occurrence of noise problems was lowered. The shooting angle is shown in Figure 2. Of the images collected in this way, only images with distinct characteristics of pests and pests were selected, and more than 25,000 images were collected. To reduce the bias of data among the collected pest classes, classes with less than 1000 images were excluded from the final dataset, and a total of 20,000 datasets were constructed and then randomly divided into 8:1:1 = training images: validation images: test images. As shown in Table 2, the number of images for the validation images and test images was unified for each class. The citrus fruit normal class covered a total of 2545 photos, citrus fruit CBC had 1716 photos, citrus leaf normal included 2455 photos, and citrus leaf CBC constituted 2545 photos. Citrus leaf Panonychus citri had a total of 9552 sheets, with 1814, and citrus leaf Toxoptera citricida a total of 1918 sheets, with 16,000 training images, 2000 validation images, and 2000 test images. All images used in the experiment were RGB images in the JPG format with a resolution of 1920 × 1090. Dataset samples are shown in Figure 3, and details of the citrus pest classes and dataset are shown in Table 2.

### 5.2. Data Transform

In this step, citrus images were simply pre-processed separately for the five models. For VGGNet, DensenNet, and EfficientNet, you need to change the image size to 224 × 224 sizes. The default input size for ViT is 224 × 224. ResNet can use various input sizes, such as 112 × 112, 224 × 224, 336 × 336, and 448 × 448. Scale the input image for all models to 224 × 224 inches. We proceed with model comparison evaluation by selecting the same input size as the remaining four models among multiple input sizes of ResNet. We applied data augmentation to training and validation datasets to solve the problem of data imbalance and model overfitting. Among the different types of data augmentation, this study uses horizontal, vertical, and rotation methods to randomly flip and rotate a 224 × 224 image dataset horizontally and horizontally by 90 degrees. The reason we used this method during data augmentation is that the above method is commonly used. Additionally, images taken in real plantations are inherently noisy. Therefore, there is a risk that the use of color jitter techniques will further spread the noise problem. So, we used basic data augmentation techniques of horizontal, vertical, and rotation methods. Since the dataset is in PIL image format, we convert it to a tensor format by applying a commonly used transform. Finally, we complete the data pre-processing by performing normalization. The test dataset is used ro verify the proposed model, and data reinforcement is not performed. It only performs image scaling, tensors, and normalization to fit model training.

### 5.3. Transfer Learning

For the network training, we performed a transfer-learning approach in which weights are transferred from a pre-trained model. Transfer learning is proven to improve the performance of the network. Scratch learning [55] shows good performance when the number of training data per class is 5000 or more. The lack of training data increases the bias of the model during training and causes it to overfit. Transfer learning can solve this problem [56]. Transfer learning reuses a model learned from a specific task to perform other tasks, and is useful in model training with a small amount of training data. In addition, when transfer learning is applied, the learning rate is faster than that of a model that learns a task from scratch, and it tends to perform better on new tasks. Transfer learning can be divided into upstream and downstream tasks. If the method of learning a specific task is called an upstream task, the learning process is called pre-training. The downstream task is a method used to solve a specific problem and has been used for image classification in this study. Transfer learning uses models trained on a wide range of datasets. In general, a model trained on the ImageNet dataset by transfer learning is used because the ImageNet dataset consists of 1000 classes and is a model that has already been trained on a large dataset. This implies that the model is suitable for extracting meaningful features from input images, and the model is trained to identify higher-level features. Therefore, models with transfer learning can be trained to be more optimized than those with scratch learning when learning new tasks.

In this study, we used a pre-trained model trained on the ImageNet dataset and then fine-tuned [57] the last layer. Fine-tuning is a method of updating a model that has completed pre-training to fit a downstream task. In this study, rather than extracting 1000 classes, the output was derived according to the number of self-collected dataset types. Since the size of the constructed dataset is large, we use the method of learning the entire model while only using the structure of the pre-training model. Through this, high accuracy and F1 score were achieved in classifying citrus pests and diseases by improving the learning speed of the model and resolving the imbalance of the collected dataset.

### 5.4. Model Training

In this section, we describe the details of the model training. The five models presented in Section 4 are divided into several versions according to their size. Among them, we trained the model by selecting VGG16, ResNet50, Densenet161, Efficient_b0, and ViT_b_16, which can be operated with ease in the experimental environment, and all models used in the experiment were pre-trained using the imagenet dataset, as described in Section 5.3 The initial learning rate used was 0.0001, and the model was trained by running 100 epochs for each model while using a batch size of 64. Models that have been trained go through a validation process, and validation proceeds by running 100 epochs per model in the same way as in training. For training and validation, the Adam optimizer, and CosineAnnealingLR were applied as the cross-entropy loss and learning rate scheduler, respectively, as mentioned above.

### 5.5. Evaluation Metrics

An evaluation of the citrus pest and disease classification model was conducted using the recall (4), precision (5), accuracy (6), and F1 score (7). true positive (TP), false negative (FN), false positive (FP), and true negative (TN) represent the predicted numbers of true positives, false negatives, false positives, and true negatives. Recall is the ratio of data predicted to be positive among actual positive data. Precision is the proportion of data that are actually positive among the data predicted as positive. Accuracy is defined as the number of correctly predicted data points divided by the total number of data points. The F1 score is the harmonic mean of precision and recall. Each model calculates the recall, precision, and accuracy values for each epoch and the f1 score based on the values.
(4)$$\begin{array}{c} Recall=\frac{TP}{TP+FN} \end{array}$$
(5)$$\begin{array}{c} Precision=\frac{TP}{TP+FP} \end{array}$$
(6)$$\begin{array}{c} Accuracy=\frac{TP+TN}{TP+TN+FP+FN} \end{array}$$
(7)$$\begin{array}{c} F1\;score=\frac{2\times Precision\times Recall}{Precision+Recall} \end{array}$$

### 5.6. Results in Validation Dataset

In this section, we evaluated the performance results of the five models obtained through the validation dataset. Table 3 described the accuracy, recall, and precision values of the five models when they achieved the highest f1 score obtained during epoch 100. In Table 3, the classification indicated by F implies fruit, and that indicated by L means leaf. The following terms indicate the type of pest, which can be seen in detail in the dataset of Table 2. In the resulting table, the class name is abbreviated for readability. All the models used in the experiment derived accuracy of 97% or more, and all f1 scores also yielded results of 96% or more. Among the five models, EfficientNet-b0 has the highest accuracy and f1 score. EfficientNet-b0 derived an accuracy of 98.8% and an f1 score of 98.2%. Even if the model’s accuracy and f1 score are high, if recall and precision are concentrated in one class, it cannot be said to have a good performance. Poor performance risks not accurately classifying citrus pests in the web application classification test, which is one of the experiments to be carried out later. As shown in Table 3, the recall and precision of the L_T.citir class are not as stable as those of other classes. The recall and precision of the ResNet50 model, showing that the DesneNet161 model is a little closer to the f1 score than the ResNet50 model, are more certain. Comparing the two models, ResNet50 used for the web application server has better performance. Among the five model validation results, efficientNet_b0, which derived the highest f1 score, showed the most stable recall and precision. Although the recall and precision of the L_T.citir class have larger deviations than the other classes, all classes except the DenseNet161 model yielded results of over 90%. Therefore, our proposed model derives high performance from the validation dataset, so the extracted five best model weights can be used for the next section, the citrus test dataset experiment.

### 5.7. Results in Test Dataset

In this section, we evaluated the performance of the citrus test dataset using the five models that had the highest f1 score in the validation dataset experiment. In Table 4, the dataset name is implied as in Table 3. Table 4 describes the experimental results of the test dataset, and an average accuracy of 98% was derived from the experimental results. All f1 scores also yielded results of 97% or more. The accuracy and f1 score results yielded similar results for both the validation and test datasets. The model that derived the highest accuracy and f1 score among the five models is EfficientNetb0. EfficientNet_b0 obtained an accuracy of 99% and f1 score of 98.6%. EfficientNet_b0 model recall and precision also achieved high performance. In particular, the citrus fruit pest dataset shows a very high classification accuracy. The confusion matrix of the five models can be seen in Figure 4. In the confusion matrix, the row represents the actual citrus pest class and the column represents the citrus pest class predicted by the model. In Figure 4, it can be seen that the most prediction error occurs in the citrus leaf normal class. The VGGNet model predicted that 16 citrus leaf Panonychus citri image samples were citrus leaf normal. Next, 10 citrus leaf toxoptera citricida image samples were predicted to be citrus leaf normal. Six citrus leaf toxoptera citricida image samples of ResNet50 were predicted as citrus leaf normals, and seven citrus leaf toxoptera citricida image samples of DenseNet161 were also predicted as citrus leaf normals. For the models of EfficinetNet and ViT, four citrus leaf Panonychus citri image samples of EfficientNet were predicted as citrus leaf normals, and nine citrus leaf Panonychus citri image samples of the ViT model were also predicted as citrus leaf normal. From this, it can be seen that the citrus leaf Panonychus citri and citrus leaf toxoptera citricida classes are most often classified as citrus leaf normal classes. Among the confusion matrices of the five models, the EfficientNet_b0 model, which derived the highest f1 score, has the smallest number of images of the wrongly predicted class. The best model that underwent five model tests was delivered to the Python Flask web server and used to classify citrus pert diseases when the user uploads an image. In this study, the weights of the EfficientNet_b0 model, which showed the best performance, were applied to the web application, and the experiment was conducted.

### 5.8. Web Application Module

The weights of the best-trained model obtained through the model training and validation were transmitted to a web server running Python Flask, and the web server stored the transmitted model. Python Flask [58] is a web server framework for interacting with users. Flask is written in Python, and it facilitates building a web server in a very easy and simple manner. It also supports extension functions that can add application functions, as well as their implementation. Extensions exist for object relationship mappers, form validation, upload management, open authentication techniques, and several common framework-related tools. The user uses the best weights among the models learned through the network structure described in Section 3 to the web server and requests to upload an image through RESTFul API [59], which passes through the trained model to prevent citrus disease and evaluate the diseases. The operation process in the network structure and web application module can be checked again in Figure 1 in Section 3.

In the study, the EfficientNet_b0 model, which showed the best performance among the models derived through the network, described above was used in the web server. The built web application performs citrus pest classification tests by mixing the self-collected test image dataset with the dataset provided by the web page. When a user requests to upload an image, the trained model classifies the image and responds with the result. Figure 3 describes the network structure and web application module operation process. In this study, Efficient-b0, which showed the best performance in validation and testing, was applied. The user can upload the desired citrus pest image through the web UI and see the result on the far right in Figure 1. The types of data obtained from the web page include citrus leaf panonychus citri, citrus leaf toxoptera citricida, and hallabong citrus CBC similar to citrus fruit CBC. The web application system containing the web application module is simply composed of an image upload window and a classification result window so that users can use it easily. Users who use the web application system can easily upload images and view pest types through the web UI and can check types of citrus pests and control methods through this system.

## 6. Discussions and Conclusions

In this paper, a computerized approach has been proposed to detect citrus pests through deep-learning methods. We collected high-resolution citrus pest images from natural plantations to build a multivariate citrus image dataset. In order to prevent noise problems, it was photographed so that the illuminance was unified and the citrus pest pattern was clearly displayed in various conditions. There are 20,000 collected images consisting of six classes. The constructed dataset detected citrus diseases by training VGGNet, ResNet, and DenseNet models that have undergone transfer learning, which is widely used in plant disease research, and EfficinetNet and ViT models, which are in the spotlight. The five models went through training, validation, and testing, and a web application server capable of effectively classifying citrus diseases was built using the EfficientNet model that showed the best performance. At present, although many deep-learning-based plant disease classification studies have been conducted, many plantations are still using humans to manually classify citrus pests. Since this method takes a long time, it is necessary to develop an application that can be automated to classify various diseases. The web application server is accessible from anywhere in the world upon enabling, and it can be used to perform citrus disease detection anywhere. This application can effectively classify citrus pests, prevent the spread of diseases, and improve the overall citrus quality.

The method proposed in this study shows valid results in simple image classification, but does not suggest specific pest locations because it does not use a detection model. We look at the collected dataset, which consists of images that can be annotated and images that are difficult to annotate. For example, CBC disease can be placed as a bounding box. However, for the citrus Panonychus citri disease, a bounding box for a specific pest location is not possible because the pest coverage is the whole leaf. Additionally, since the size of the dataset is 20,000 sheets, it is difficult to annotate them manually. For this reason, we did not comment on the dataset we built. This problem arose because the actual farm contained more citrus pests and we wanted to collect images that could work seamlessly with the tin, but we only collected data for 2021 from a specific region. However, the dataset used in the experiment includes rare disease types such as citrus Toxoptera citricida, including the CBC disease group that was widely used in previous studies. A challenging part of deep learning research is collecting large datasets covering different kinds of diseases. This is because relatively small datasets overfit the model. The dataset used in this study is a large dataset of 20,000 sheets and contains unfamiliar citrus pest types. In addition, current citrus disease recognition research has limitations in data collection, focusing on unpublished data, paid data, and the Kaggle dataset. We make our dataset public with this study. By applying this dataset to existing citrus disease recognition studies, we can classify more types of citrus diseases. Further annotations to this dataset in the future can pinpoint specific locations of disease in detection models such as YOLO [60] and RefineDet [61]. Additionally, although pests appear on particular parts of fruits and leaves, the scope of diseases such as the citrus Panonychus citri is holistic. Therefore, data pre-processing studies that can better represent disease patterns should be performed. In the future, if the above studies are carried out, it will be possible to generalize to studies applicable to other varieties beyond citrus in actual farmland.

## Figures and Tables

**Figure 1 sensors-22-08911-f001:**
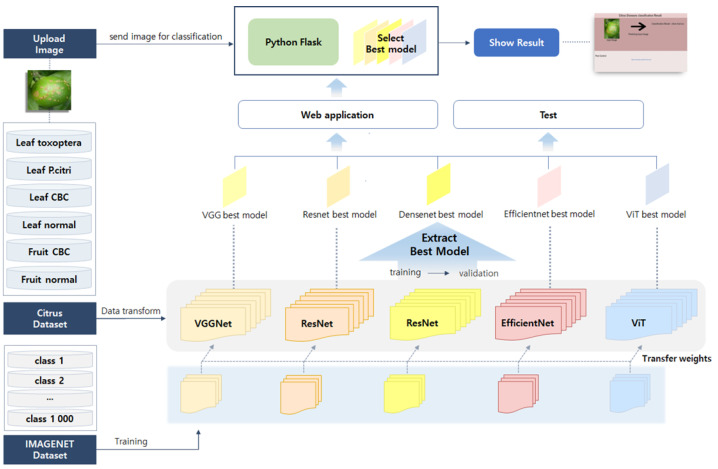
Citrus pest classification system network architecture.

**Figure 2 sensors-22-08911-f002:**
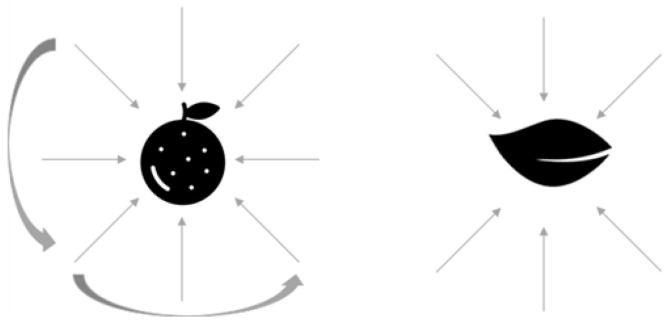
Dataset shooting angle.

**Figure 3 sensors-22-08911-f003:**
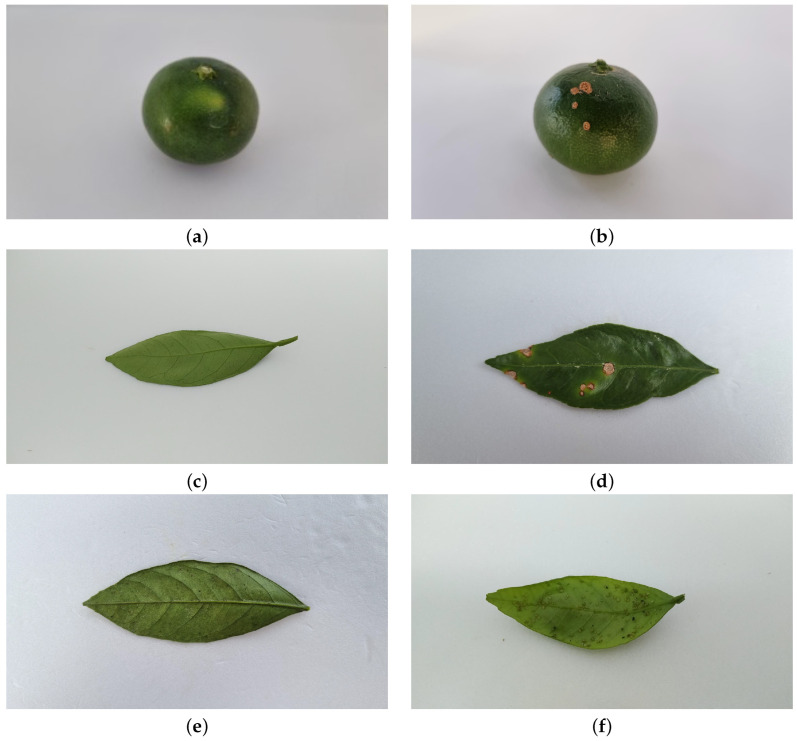
*Citrus unshiu* pest disease image sample: (**a**) Citrus fruit normal. (**b**) citrus fruit CBC. (**c**) citrus leaf normal. (**d**) citrus leaf CBC. (**e**) citrus leaf Panonychus citri. (**f**) citrus leaf Toxoptera citricida.

**Figure 4 sensors-22-08911-f004:**
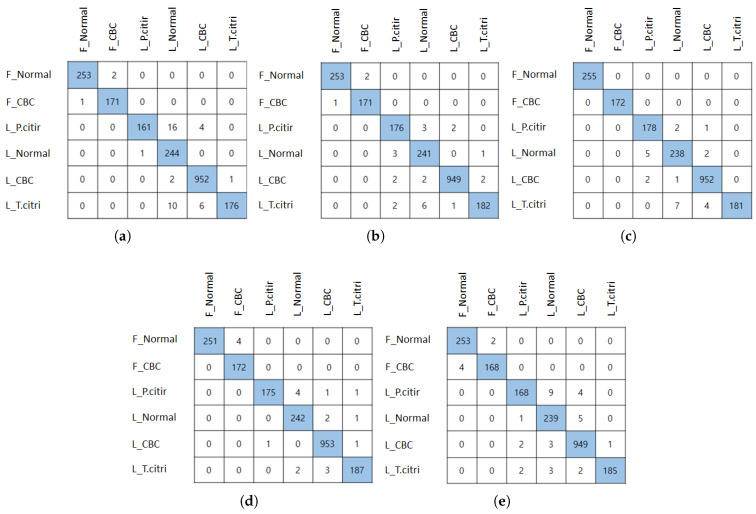
*Confusion matrix of test dataset* (**a**) VGGNet16. (**b**) ResNet50. (**c**) DenseNet161. (**d**) EfficinetNet_b0. (**e**) ViT_b_16.

**Table 1 sensors-22-08911-t001:** Comparison of five model parameters.

Model Name	Parameters
VGGNet16	134,285,126
ResNet50	23,520,326
DenseNet161	26,485,254
ViT_b_16	85,803,270
EfficientNet_b0	4,015,234

**Table 2 sensors-22-08911-t002:** Summary of the dataset information.

Citrus Pest Disease Type	Training Images	Validation Images	Test Images	Total Images	Image Size
Citrus Fruit Normal	2034	225	255	2545	1920 × 1080
Citrus Fruit CBC	1372	172	172	1716	1920 × 1080
Citrus Leaf Normal	1965	245	245	2455	1920 × 1080
Citrus Leaf CBC	7642	955	955	9552	1920 × 1080
Citrus Leaf Panonychus citri	1452	181	181	1814	1920 × 1080
Citrus Leaf Toxoptera citricida	1534	192	192	1918	1920 × 1080
Total	16,000	2000	2000	20,000	

**Table 3 sensors-22-08911-t003:** Five model validation classification performance of the citrus dataset.

Model	Accuracy	F1 Score	Class	Parameters	Parameters
VGG16	0.977	0.967	F_Normal	0.985	0.982
			F_CBC	0.974	0.981
			L_Normal	0.948	0.971
			L_CBC	0.989	0.999
			L_P.citri	0.952	0.946
			L_T.citri	0.969	0.90
			macro avg	0.97	0.97
			weighted avg	0.98	0.98
ResNet50	0.983	0.976	F_Normal	0.982	0.987
			F_CBC	0.983	0.974
			L_Normal	0.972	0.962
			L_CBC	0.999	0.993
			L_P.citri	0.974	0.964
			L_T.citri	0.938	0.99
			macro avg	0.97	0.97
			weighted avg	0.98	0.98
DenseNet161	0.984	0.977	F_Normal	0.986	0.986
			F_CBC	0.984	0.984
			L_Normal	0.971	0.964
			L_CBC	0.998	0.987
			L_P.citri	0.953	0.971
			L_T.citri	0.885	1.0
			macro avg	0.97	0.97
			weighted avg	0.98	0.98
EfficientNet	0.988	0.982	F_Normal	0.995	0.992
			F_CBC	0.989	0.992
			L_Normal	0.995	0.992
			L_CBC	0.997	0.998
			L_P.citri	0.996	0.947
			L_T.citri	0.925	1.0
			macro avg	0.97	0.98
			weighted avg	0.98	0.98
ViT	0.972	0.961	F_Normal	0.974	0.974
			F_CBC	0.956	0.956
			L_Normal	0.963	0.942
			L_CBC	0.996	0.982
			L_P.citri	0.933	0.955
			L_T.citri	0.914	0.988
			macro avg	0.97	0.98
			weighted avg	0.98	0.98

**Table 4 sensors-22-08911-t004:** Five model test classification performance of the citrus dataset.

Model	Accuracy	F1 Score	Class	Parameters	Parameters
VGG16	0.979	0.97	F_Normal	0.992	0.996
			F_CBC	0.994	0.988
			L_Normal	0.996	0.897
			L_CBC	0.997	0.99
			L_P.citri	0.969	0.89
			L_T.citri	0.917	0.994
			macro avg	0.96	0.98
			weighted avg	0.98	0.98
ResNet50	0.986	0.98	F_Normal	0.992	0.996
			F_CBC	0.994	0.983
			L_Normal	0.956	0.984
			L_CBC	0.994	0.997
			L_P.citri	0.972	0.962
			L_T.citri	0.948	0.984
			macro avg	0.98	0.98
			weighted avg	0.99	0.99
DenseNet161	0.985	0.977	F_Normal	1.0	1.0
			F_CBC	1.0	1.0
			L_Normal	0.971	0.96
			L_CBC	0.997	0.993
			L_P.citri	0.983	0.962
			L_T.citri	1.0	0.943
			macro avg	0.98	0.99
			weighted avg	0.99	0.99
EfficientNet	0.99	0.986	F_Normal	0.984	1.0
			F_CBC	1.0	0.977
			L_Normal	0.988	0.976
			L_CBC	0.998	0.994
			L_P.citri	0.967	0.994
			L_T.citri	0.974	0.984
			macro avg	0.99	0.99
			weighted avg	0.99	0.99
ViT	0.981	0.975	F_Normal	0.992	0.984
			F_CBC	0.977	0.988
			L_Normal	0.976	0.941
			L_CBC	0.994	0.989
			L_P.citri	0.928	0.971
			L_T.citri	0.964	0.995
			macro avg	0.97	0.98
			weighted avg	0.98	0.98

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
