# Peer review of "Automatic Classification Service System for Citrus Pest Recognition Based on Deep Learning"

_sensors, 2022, doi:10.3390/s22228911_

Round 1

Reviewer 1 Report

This paper proposes an automatic citrus disease classification service system by using five pre-trained deep learning models to recognize citrus diseases and transmits them to a web application server. The paper is well written, however, it needs to be improved as follows:

Comment 1: The merit of the proposed approach is supported by the results, but I miss on the paper a bit more discussion on why these techniques were chosen for this problem and had not been considered before.

 Comment 2: The evaluation of related works is weak. Section II. Some of the work discussed in Deep learning-based fruit crop disease research is outdated. Research related to plant disease recognition using the latest technology should be added.

Comment 3: Describe in more detail the data pre-processing steps. The plant images taken in real-world conditions on-site usually have noisiness issues. How did you deal with denoising and data set imbalance issues? Please discuss in detail.

Comment 4: Why was training discontinued after 100 epochs? How do you avoid/prevent overfitting

Comment 5: Authors should further clarify and elaborate novelty in their contribution. What are the limitations of the present work, and What are the practical implications of this research?

Author Response

Answers to Reviewer's Comments

Manuscript Title: Automatic Classification Service System for Citrus Pest Recognition based on Deep Learning

First of all, we would like to appreciate your careful and detailed comments on the manuscript listed above.

Answers to your specific comments are as follows,

This paper proposes an automatic citrus disease classification service system by using five pre-trained deep learning models to recognize citrus diseases and transmits them to a web application server. The paper is well written, however, it needs to be improved as follows:

Comment 1: The merit of the proposed approach is supported by the results, but I miss on the paper a bit more discussion on why these techniques were chosen for this problem and had not been considered before.

Answer 1: We added a contribution at the end of the introduction. Existing studies use machine learning and deep learning methods to recognize and classify citrus pests. As machine learning methods operate based on user-defined features, image feature values might be missing in image classification. To solve these problems, deep learning technology has been used in recent plant disease diagnosis research. Deep learning methods automatically extract features from images and produce more accurate results than machine learning methods, but when the data size is small, overfitting problems occur. The problem with the current study is that the citrus disease data set is small. So, we created a citrus disease data set and built and tested it as a real citrus disease recognition service system.

(line 80, Section I, page 2)

We have developed a novel citrus pest dataset comprising six disease detection classes. The constructed dataset consists citrus images that are either infected or non-infected by pests in Jeju Island, South Korea, in 2021. The constructed dataset provides a total of 20,000 high-quality images with a resolution of 1920×1090. Currently, Citrus Open Datasets are either low-resolution or paid. We publish the data sets used in the study free of charge https://github.com/LeeSaeBom/citrus. A detailed description of the dataset is provided in Section V.

(line 92, Section I, page 3)

Application development is required to automate the classification of various diseases. The web application server has the advantage that it can be accessed from anywhere in the world where the Internet is available. In most citrus cultivation sites, workers manually determine the presence or absence of pests and classify disease types. It is difficult for non-professional workers to quickly determine the type of pest. Based on these problems, we developed our own web application system, and using it, non-professional workers can easily determine the pests and diseases.      

Comment 2: The evaluation of related works is weak. Section II. Some of the work discussed in Deep learning-based fruit crop disease research is outdated. Research related to plant disease recognition using the latest technology should be added.

Answer 2: Fruit crop disease recognition research uses statistical, machine learning, and deep learning methods. We accepted the reviewer's corrections and added research articles related to plant disease recognition that was published within the last 3 years. Some of the existing related research technologies have been changed to the latest technologies.

(line 155, Section II, page 4)

Syed Ab Rahman et al. detected citrus disease using a two-step deep convolutional neural network based on Faster R-CNN, and the network structure is a feature extractor, region proposal network (RPN), region of interest (ROI) pooling, and it consists of four components of a classifier. This study has the advantage of fast training speed and memory saving.

Comment 3: Describe in more detail the data pre-processing steps. The plant images taken in real-world conditions on-site usually have noisiness issues. How did you deal with denoising and data set imbalance issues? Please discuss in detail.

Answer 3: Image data obtained under real-world conditions usually suffers from noise issues. Added information about data collection environment, lighting, exposure level, background inconsistency, and how to remove noise.

(line 332, Section V, page 8)

First, the exact screen composition and ratio were set so that the entire subject came out. After that, the height, angle, distance, and lighting distance were adjusted for each pest type for consistent image quality. Also, only a single breed was photographed on a white background, and the photograph was taken so that the subject's shaking, shading, and light reflection did not occur at the time of the shooting. Finally, the pictures were taken so that the unique characteristics of each breed, such as color and pattern, were clearly visible. High-quality images were collected using this method. Therefore, the rate of occurrence of noise problems is lowered. The shooting angle is shown in Figure 2. 

Comment 4: Why was training discontinued after 100 epochs? How do you avoid/prevent overfitting?

Answer 4: There were no significant results from training when the model exceeded 100 reps. Rather, the f1 score decreases right after 100th epoch. And data augmentation was applied to prevent overfitting. Please see the added content to Section V. Data Transformer.

 (line 358, Section V, page 10)

We applied data augmentation to the training and validation datasets to solve the data imbalance problem and model overfitting issue.

Comment 5: Authors should further clarify and elaborate novelty in their contribution. What are the limitations of the present work, and What are the practical implications of this research?

Answer 5: We describe in more detail the limitations of our method. The proposed method showed effective results with simple image classification but failed to suggest a specific pest area. Guidance is needed to present detailed areas using future detection models. However, we developed a web application based on a deep learning model by building a new data set. Using the method suggested by natural farmland will be helpful for ordinary people or farmers who have difficulty distinguishing citrus pests.

 (line 499, Section VI, page 15)

However, the dataset used in the experiment contains rare disease types such as citrus Toxoptera citricida, including the CBC disease class that has been widely used in previous studies. A very difficult part of deep learning research is collecting large datasets containing various kinds of diseases. This is because a relatively small dataset leads to hardships in model training. The data set used in this study contains unfamiliar citrus pests and will be made public in this study. Using this data set for existing studies, more types of citrus diseases can be recognized. However, the dataset used in this study was not annotated. Therefore, it is necessary to build an annotated dataset in the future. Utilizing detection models such as YOLO and RefineDet could detect specific disease areas.

Again, we appreciate your careful and detailed comments on this manuscript, which were very helpful in revising.

Thank you very much.

Reviewer 2 Report

This paper proposes an automatic citrus disease classification service system by using five pre-trained deep learning models to recognize citrus diseases and transmits them to a web application server. The paper is well written, however, it needs to be improved as follows:

Comment 1:The motivation for choosing five deep learning models among the contributions described in the last paragraph of the introduction section should be more clearly explained.

Comment 2: Plant disease recognition research also makes heavy use of machine learning methods. What was the key motivation behind focusing on the Deep Learning? 

Comment 3: Another dataset would consolidate the work if the authors obtain a consisting
result.

Comment 4: In Section V. Transfer Learning, fine-tuning methods include a method of newly learning the whole model, a method of freezing a part of the conv base, and a method of freezing the conv base and newly learning only the classifier. Additional information is needed on which strategies were used in this study.

Author Response

Answers to Reviewer's Comments

Manuscript Title: Automatic Classification Service System for Citrus Pest Recognition based on Deep Learning

First of all, we would like to appreciate your careful and detailed comments on the manuscript listed above.

Answers to your specific comments are as follows,

This paper proposes an automatic citrus disease classification service system by using five pre-trained deep learning models to recognize citrus diseases and transmits them to a web application server. The paper is well written, however, it needs to be improved as follows:

Comment 1: The motivation for choosing five deep learning models among the contribution described in the last paragraph of the introduction section should be more clearly explained.

Answer 1: We added motivation for choosing five deep learning models to contribution in the last paragraph of the introduction section.

(line 87, Section I, page 2)

We use EfficientNet and ViT models, which are the latest algorithms in this area, including VGGNet, ResNet, and DenseNet models that are commonly used for the classification and detection of plant pests and diseases. All five models can use the pre-training method, and high accuracy and f1 score derivation are possible. VGGNet, ResNet, DenseNet and EfficinetNet models can extract local features of the feature map using a convolution layer, and the ViT model uses a transformer, so global features of the feature map can be extracted

Comment 2: Plant disease recognition research also makes heavy use of machine learning methods. What was the key motivation behind focusing on the Deep Learning?

Answer 2: A representative method among machine learning-based plant recognition research is to use the principal component analysis (PCA) technique. The PCA method has the strength to find components that are judged to explain the given data well. However, as statistics and machine learning detect plant diseases based on self-generated image features, important information in the images may be lost. This, along with poor model performance, leads to difficulties in detecting plant pests immediately. Deep learning methods solve problems in machine learning by going through layers and using the features of the image determined by the layers. The downside of deep learning models is that they take longer than machine learning methods. However, with the advancement of computing systems, the training time of deep learning-based models is decreasing. Therefore, deep learning techniques are more effective than machine learning in plant disease recognition research.

(line 124, Section II, page 3)

Another representative method among machine learning-based infected plant recognition research is to use principal component analysis (PCA). The PCA method has the advantage of finding components that are judged to explain the given data well. However, as statistics and machine learning detect plant diseases based on self-generated image features, important information in the images may be lost. This, along with poor model performance, leads to difficulties in immediately detecting plant pests. Deep learning methods solve problems in machine learning by going through layers and using features of an image determined by the layers. The downside of deep learning models is that they take longer than machine learning methods. However, with the advancement of computing systems, the training time of deep learning-based models is decreasing. Therefore, deep learning techniques are more effective than machine learning in plant disease recognition research. In order to solve this problem over the past 5 years, research combining machine learning and deep learning has appeared, and deep network learning has become possible with the development of GPUs, and deep learning-based crop disease classification research is in progress has been active.

Comment 3: Another dataset would consolidate the work if the authors obtain a consisting
result.

Answer 3: When selection data, we performed third-party quality verification by external experts. Cross-validation was not carried out because the inspected test data were prepared separately.

Comment 4: In Section V. Transfer Learning, fine-tuning methods include a method of newly learning the whole model, a method of freezing a part of the conv base, and a method of freezing the conv base and newly learning only the classifier. Additional information is needed on which strategies were used in this study.

Answer 4: We used the method of training the model all new to fit our dataset while only using the structure of the pre-trained model.

(line 385, Section V, page 10)

 Fine-tuning is a method of updating a model that has completed pre-training to fit a downstream task. In this study, rather than extracting 1,000 classes, the output was derived according to the number of self-collected data set types. Since the size of the constructed dataset is large, we use the method of learning the entire model while only using the structure of the pre-training model. Through this, high accuracy and F1 score were achieved in classifying citrus pests and diseases by improving the learning speed of the model and resolving the imbalance of the collected dataset.

Again, we appreciate your careful and detailed comments on this manuscript, which were very helpful in revising.

Thank you very much.

Reviewer 3 Report

In order to identify citrus diseases and send them to a web application server, this study offers an autonomous citrus disease classification service system using five pre-trained deep learning models. Although the work is nicely written, the following improvements should be made:  

Comment 1: The identified problem of the article is that there is "a need for the latest AI-based tools that can predict the prognosis of diseases by detecting pests in advance" but in the discussion, it says the results do "not suggest a specific pest area." Therefore, the results do not address the article's question.

Comment 2: The novelty of the study is not clear. The EfficientNet deep learning model architecture and its various modifications and optimizations have already been used before in various studies including for recognition of retinal diseases. The authors must explicitly state their innovation in the research field and difference from previous works. 

Comment 3: Describe in more detail the data pre-processing steps. The plant images taken. in real-world conditions on-site usually have noisiness issues. How did you deal with denoising and data set imbalance issues? Please discuss in detail. Section III data transformation technology of network architecture is poor. You should describe how you transformed the data and why you chose the input image size.

Comment 4: Fig.4 Make the resolution of the image clear. Labels in the confusion matrix.  are not clearly visible.

Comment 5: The author should make the discussion section a bit more clear. Discuss the methodology and limitations of this study. In the "real world," how practical is it to take pictures of pests that occur in actual agricultural settings? 

Author Response

Answers to Reviewer's Comments

Manuscript Title: Automatic Classification Service System for Citrus Pest Recognition based on Deep Learning

First of all, we would like to appreciate your careful and detailed comments on the manuscript listed above.

Answers to your specific comments are as follows,

This paper proposes an automatic citrus disease classification service system by using five pre-trained deep learning models to recognize citrus diseases and transmits them to a web application server. The paper is well written, however, it needs to be improved as follows:

Comment 1,5:
(a) The identified problem of the article is that there is "a need for the latest AI-based tools that can predict the prognosis of diseases by detecting pests in advance" but in the discussion, it says the results do "not suggest a specific pest area." Therefore, the results do not address the article's question.
(b) The author should make the discussion section a bit more clear. Discuss the methodology and limitations of this study. In the "real world," how practical is it to take pictures of pests that occur in actual agricultural settings?

Answer 1,5:

(a) We took the comments of our reviewers and modified the discussion section to clearly state what the research was trying to say. What we are trying to say in this paper is the need for a service in which an AI-based tool automatically classifies pests rather than manually classifying pests by humans. We derived f1-scores of over 97% for all five models on the citrus pest dataset through experiments. Furthermore, we built a web application system based on the experiment in order to make the research usable by people who are actually engaged in agriculture. Currently, the most common task for identifying diseases in cultivated fields is to manually check for pests. This method is time-consuming and difficult for non-experts. If you use the method we propose, you can quickly and easily identify the disease.

The discussion section describes the contributions and limitations of the study. In the reviewer discussion part, the meaning of “do not suggest a specific pest area” means that the research proposed in the paper is simply a service to classify diseases, so further research is needed on models that detect specific locations of diseases in the future. The reason that the above work was not performed in the proposed study is that it is difficult to perform manual data annotation because the dataset consists of 20,000 samples. Therefore, in the discussion section, addressing the need to annotate data to determine the location of specific pests, and conducting disease detection studies using annotated datasets can suggest better plant disease, recognition models.

(b) It is very important to collect image data of pests that occur in real agriculture. First, the actual agricultural population is decreasing every year due to the decline and aging of the agricultural population. Therefore, we need to study how to recognize the disease with minimal human intervention. Second, due to the nature of citrus trees, trees infected with pests are highly contagious and must be removed unconditionally. A quick determination of whether a tree is infected with pests will affect citrus yields. Finally, although the plant data set has published images of plant diseases, the plant disease dataset for research is still lacking. And so far, studies that have done citrus classification studies use unpublished or paid data. Therefore, it is necessary to collect a sufficient amount of data to carry out various and many studies. Also, the more pictures of pests encountered in real agricultural environments, the easier it is to apply the findings to arable land.

(line 498, Section VI, page 15)

The method proposed in this study shows valid results in simple image classification but does not suggest specific pest locations because it does not use a detection model. Looking at the collected dataset consists of images that can be annotated and images that are difficult to annotate. For example, CBC disease can be placed as a bounding box. However, for the citrus Panonychus citri disease, a bounding box for a specific pest location is not possible because the pest coverage is the whole leaf. And since the size of the dataset is 20,000 sheets, it is difficult to annotate them manually. For this reason, we did not comment on the dataset we built. This problem arose because the actual farm contained more citrus pests and we wanted to collect images that could work seamlessly with the tin, but we only collected data for 2021 from a specific region. However, the dataset used in the experiment includes rare disease types such as citrus Toxoptera citricida, including the CBC disease group that was widely used in previous studies. A challenging part of deep learning research is collecting large datasets covering different kinds of diseases. This is because relatively small datasets overfit the model. The dataset used in this study is a large dataset of 20,000 sheets and contains unfamiliar citrus pest types. In addition, current citrus disease recognition research has limitations in data collection, focusing on unpublished data, paid data, and the Kaggle dataset. We make our dataset public with this study. By applying this dataset to existing citrus disease recognition studies, we can classify more types of citrus diseases. Further annotations to this dataset in the future can pinpoint specific locations of disease in detection models such as YOLO and RefineDet. Also, although pests appear on particular parts of fruits and leaves, the scope of diseases such as the citrus Panonychus citri is holistic. Therefore, data preprocessing studies that can better represent disease patterns should be performed. In the future, if the above studies are carried out, it will be possible to generalize to studies applicable to other varieties beyond citrus in actual farmland.

Comment 2: The novelty of the study is not clear. The EfficientNet deep learning model architecture and its various modifications and optimizations have already been used before in various studies including for recognition of retinal diseases. The authors must explicitly state their innovation in the research field and difference from previous works.

Answer 2: We added a novelty of the study in Section I. Deep learning research on fruit crop disease recognition is being conducted in various ways. Several CNN-based models, such as GoogleNet, ResNet, and EfficientNet, including VGGNet, which are the most representative CNN models, are being used for fruit crop disease recognition problems. However, since most of the currently conducted fruit crop recognition studies use public datasets such as Kaggle and plantVillage, the diversity of datasets is insufficient. In particular, amongruit crops, citrus is an item that is produced and cultivated worldwide. Citrus crops are very sensitive to weather conditions and prone to disease. If a citrus tree is infested with pests, the citrus tree must be removed. Therefore, it is necessary to predict the disease in advance through citrus disease recognition studies. However, as mentioned above, there are still difficulties in citrus pest classification studies due to the lack of citrus datasets. This study does not focus on the architecture of the EfficientNet model and various modifications and optimization methods. In the real field, we collect high-resolution citrus disease data images for a year to build a dataset and proceed with citrus disease recognition using five deep learning models. Among the five models, the EfficientNet model showed the best performance, so a web application was developed using the weights of the EfficientNet model. Web applications can be accessed from anywhere in the world where the Internet is available. Using this, we believe that citrus workers in the real environment will be able to help classify citrus diseases easily.

 (line 80, Section I, page 2)

We have developed a novel citrus pest dataset comprising six disease detection classes. The constructed dataset consists citrus images that are either infected or non-infected by pests in Jeju Island, South Korea, in 2021. The constructed dataset provides a total of 20,000 high-quality images with a resolution of 1920×1090. Currently, Citrus Open Datasets are either low-resolution or paid. We publish the data sets used in the study free of charge https://github.com/LeeSaeBom/citrus. A detailed description of the dataset is provided in Section V.

(line 87, Section I, page 2)

We use EfficientNet and ViT models, which are the latest algorithms in this area, including VGGNet, ResNet, and DenseNet models that are commonly used for the classification and detection of plant pests and diseases \cite{b21, b22, b23}. All five models can use the pre-training method, and high accuracy and f1 score derivation are possible. VGGNet, ResNet, DenseNet and EfficinetNet models can extract local features of the feature map using a convolution layer, and the ViT model uses a transformer, so global features of the feature map can be extracted.

Comment 3: Describe in more detail the data pre-processing steps. The plant images taken. in real-world conditions on-site usually have noisiness issues. How did you deal with denoising and data set imbalance issues? Please discuss in detail. Section III data transformation technology of network architecture is poor. You should describe how you transformed the data and why you chose the input image size.

Answer 3: Image data obtained under real-world conditions usually suffer from noise issues. Added information about data collection environment, lighting, exposure level, background inconsistency, and how to remove noise. And added content to Section III data transformation.

(line 333, Section V, page 8)             

First, the exact screen composition and ratio were set so that the entire subject came out. After that, the height, angle, distance, and lighting distance were adjusted for each pest type for consistent image quality. Also, only a single breed was photographed on a white background, and the photograph was taken so that the subject's shaking, shading, and light reflection did not occur at the time of the shooting. Finally, the pictures were taken so that the unique characteristics of each breed, such as color and pattern, were clearly visible. High-quality images were collected using this method. Therefore, the rate of occurrence of noise problems is lowered. The shooting angle is shown Figure 2.

(line 181, Section III, page 4)

In the data transformation step, model training was conducted after image resizing at 224 × 224 and stochastically rotating the image vertically or horizontally. The detailed data transformation process is described in Section V.

(line 354, Section V, page 9)

In this step, citrus images were simply preprocessed separately for the five models. For VGGNet, DensenNet, and EfficientNet, you need to change the image size to 224 × 224 sizes. The default input size for ViT is 224 × 224. ResNet can use various input sizes such as 112 × 112, 224 × 224, 336 × 336, and 448 × 448. Scale the input image for all models to 224 × 224 inches. We proceed with model comparison evaluation by selecting the same input size as the remaining 4 models among multiple input sizes of ResNet. We applied data augmentation to training and validation datasets to solve the problem of data imbalance and model overfitting. Among the different types of data augmentation, this study uses horizontal, vertical, and rotation methods to randomly flip, rotate, and rotate a 224 × 224 image dataset horizontally and horizontally by 90 degrees. The reason we used this method during data augmentation is that the above method is commonly used a lot. Also, images taken in real plantations are inherently noisy. Therefore, there is a risk that the use of color jitter techniques will further spread the noise problem. So, we used basic data augmentation techniques of horizontal, vertical, and rotation methods. Since the dataset is in PIL image format, we convert it to a tensor format by applying a commonly used transform. Finally, we complete the data preprocessing by performing normalization. The test dataset is for verifying the proposed model, and data reinforcement is not performed. It only performs image scaling, tensors, and normalization to fit model training.

Comment 4: Fig.4 Make the resolution of the image clear. Labels in the confusion matrix.  are not clearly visible.

Answer 4: In Figure 4, we increased the resolution of the figure and increased the size of the class names in the errata.

(line 456, Section V, page 14)

In Figure 4, it can be seen the most prediction error occurs in the citrus leaf normal class.

Again, we appreciate your careful and detailed comments on this manuscript, which were very helpful in revising.

Thank you very much.
